# Adaptation of the H7N2 Feline Influenza Virus to Human Respiratory Cell Culture

**DOI:** 10.3390/v14051091

**Published:** 2022-05-19

**Authors:** Wataru Sekine, Akiko Takenaka-Uema, Haruhiko Kamiki, Hiroho Ishida, Hiromichi Matsugo, Shin Murakami, Taisuke Horimoto

**Affiliations:** Laboratory of Veterinary Microbiology, Graduate School of Agricultural and Life Sciences, The University of Tokyo, Tokyo 113-8657, Japan; wataru-sekine@g.ecc.u-tokyo.ac.jp (W.S.); atakiko@g.ecc.u-tokyo.ac.jp (A.T.-U.); asotus2155@gmail.com (H.K.); i.hiroho417@gmail.com (H.I.); tera_cyurin@yahoo.co.jp (H.M.); shin-murakami@g.ecc.u-tokyo.ac.jp (S.M.)

**Keywords:** feline influenza, H7N2, hemagglutinin, neuraminidase, mutation, adaptation

## Abstract

During 2016–2017, the H7N2 feline influenza virus infected more than 500 cats in animal shelters in New York, USA. A veterinarian who had treated the cats became infected with this feline virus and showed mild respiratory symptoms. This suggests that the H7N2 feline influenza virus may evolve into a novel pandemic virus with a high pathogenicity and transmissibility as a result of mutations in humans. In this study, to gain insight into the molecular basis of the transmission of the feline virus to humans, we selected mutant viruses with enhanced growth in human respiratory A549 cells via successive passages of the virus and found almost all mutations to be in the envelope glycoproteins, such as hemagglutinin (HA) and neuraminidase (NA). The reverse genetics approach revealed that the HA mutations, HA1-H16Q, HA2-I47T, or HA2-Y119H, in the stalk region can lead to a high growth of mutant viruses in A549 cells, possibly by changing the pH threshold for membrane fusion. Furthermore, NA mutation, I28S/L, or three-amino-acid deletion in the transmembrane region can enhance viral growth in A549 cells, possibly by changing the HA–NA functional balance. These findings suggest that the H7N2 feline influenza virus has the potential to become a human pathogen by adapting to human respiratory cells, owing to the synergistic biological effect of the mutations in its envelope glycoproteins.

## 1. Introduction

Influenza A viruses (IAVs) are endemic in humans and other mammals [1]. As the natural reservoir of IAV is a wild aquatic bird, the virus spreads worldwide and can acquire the ability to cross the species barrier from birds to mammals by chance [2,3]. From December 2016 to February 2017, an IAV outbreak with mainly respiratory signs, such as coughing and nasal mucus discharge, occurred in more than 500 cats in animal shelters in New York, USA, and it was found that it was caused by the virus derived from H7N2 low-pathogenicity avian IAV, which was closely related to that circulating in live bird markets between the late 1990s and the early 2000s [4,5]. Two shelter workers were confirmed to be infected with the virus, including a veterinarian who had treated cats, became infected with the virus, and displayed a mild respiratory illness [6,7].

Several IAV infections have been reported in companion animals, such as dogs and cats [8], including H5 highly pathogenicity avian IAVs, causing severe disease and death in some cases [9,10,11,12,13]. In addition, infections with human IAVs, likely transmitted from pet owners, have been reported in companion animals [14,15]. Although the transmission of IAV from companion animals to humans has not been reported extensively, this observation suggests that they may become intermediate hosts for avian IAVs to be transmitted to humans through mutations or reassortment with human IAV, because of their high chance of contact with humans.

In the established feature with avian IAVs of the H5 or H7 subtype, H7N2 low-pathogenicity avian IAV possess the potential to convert to a highly pathogenicity form by acquiring mutations at the hemagglutinin (HA) cleavage site [16]. In addition, human infections with H7 avian viruses, including the H7N2 virus, have been reported, albeit in limited cases. Among them, it is well documented that, in 2013–2017, H7N9 low-pathogenicity avian IAV and its later highly pathogenicity form were transmitted from poultry to humans, resulting in a large number of human infections and deaths in China, posing a pandemic threat [17]. Therefore, attention needs to be paid to the emergence of mutant viruses from H7N2 feline IAV, leading to a high pathogenicity in cats and transmissibility to humans. In this study, to gain insight into the molecular basis of the transmission of the H7N2 feline virus to humans, we investigated mutations that could induce enhanced growth in human respiratory cell culture.

## 2. Materials and Methods

### 2.1. Cells and Viruses

Madin-Darby canine kidney (MDCK) cells were acquired from American Type Culture Collection (ATCC; CCL-34). These cells were cultured in Eagle’s Minimum Essential Medium (MEM, Life Technologies, Tokyo, Japan) containing 5% newborn calf serum (NCS). We acquired human embryonic kidney (HEK293T) cells from the RIKEN BioResource Research Center (RCB2202) and obtained human lung adenocarcinoma epithelial (A549) cells from JCRB Cell Bank, National Institutes of Biomedical Innovation, Health, and Nutrition (JCRB0076); both cell lines were cultivated in Dulbecco’s Modified Eagle Medium (DMEM) containing 10% fetal bovine serum (FBS). All of the cells were incubated in 5% CO_2_ at 37 °C. A/feline/New York/WVDL-14/2016 (H7N2) [18], kindly provided by Dr. Toohey-Kurth (University of Wisconsin) and Dr. Kawaoka (University of Tokyo), was propagated in MDCK cells.

### 2.2. Passages of H7N2 Feline Influenza Virus in A549 Cells

A549 cells in 6-well plates were infected with the feline virus at a multiplicity of infection (MOI) of 0.01, and were subsequently maintained in DMEM supplemented with 0.3% bovine serum albumin (DMEM/BSA) and 0.5 µg/mL tosylsulfonyl phenylalanyl chloromethyl ketone (TPCK)-trypsin (Worthington, Lakewood, NJ, USA). After incubation for 72 h, the supernatants were collected and centrifuged to remove the cell debris. Afterwards, fresh A549 cells were infected with the feline virus under the same conditions. This process was repeated 16 times, and three lines of A549-adapted viruses were obtained (designated as Aad-1, -2, and -3).

### 2.3. Growth Kinetics of the Virus in A549 Cells

A549 cells were inoculated with viruses at an MOI of 0.01 and were maintained in DMEM/BSA with 0.5 µg/mL TPCK-trypsin. Supernatants were collected at 0, 12, 24, 36, 48, and 60 h post infection and subjected to virus titration in MDCK cells using a plaque assay. For the plaque assay, confluent monolayers of MDCK cells were washed with MEM/BSA and inoculated with the diluted virus. Afterwards, the cells were incubated for 60 min at 37 °C. Thereafter, the cells were washed to remove the virus inoculum and then overlaid with MEM/BSA containing 1% agarose and 1 μg/mL TPCK-trypsin. The cell plates were incubated at 37 °C for 48 h. After incubation, we counted the number of virus plaques that were formed in the cell monolayers.

### 2.4. Sequence Analysis

Viral RNA extraction from the supernatant of the virus-infected cells was performed using ISOGEN-LS (Nippon Gene, Tokyo, Japan). The entire viral cDNA was generated by Uni12 primer using ReverTra Ace (TOYOBO, Osaka, Japan), and each genome segment was amplified from cDNA by PCR using specific primers [19]. The PCR products were purified using a Fast Gene Gel/PCR Extraction Kit (NIPPON Genetics, Tokyo, Japan). The BigDye Terminator (v3.1) sequencing kit and an automated sequencer (Life Technologies, Japan, Applied Biosystems 3130xl, Tokyo, Japan) were used to sequence the purified PCR products. Sequences were assembled using the ATGC software version 15 (GENETYX, Tokyo, Japan). Thereafter, the assembled sequences were compared with those of the wild-type and Aad viruses.

### 2.5. Reverse Genetics

Plasmid-based reverse genetics was performed as previously described [20]. Briefly, eight viral RNA-synthesis pHH21-based plasmids, which were constructed from wild-type, adapted, and mutant viruses, separately, and four protein-expression (PB2, PB1, PA, and NP) pCAGGS-based plasmids, which were prepared from A/WSN/33 (H1N1) [20], were mixed with the transfection reagent, TransIT-293T (Mirus, Madison, WI, USA), and incubated at 23 °C for 15 min. The mixture was added to HEK293T cells and the cells were cultured in Opti-MEM (Life Technologies/Gibco, Grand Island, NY, USA). At 48 h post transfection, the supernatants were harvested and added to the MDCK cell culture. The cells were then maintained in MEM/BSA with 1 µg/mL TPCK-trypsin for virus propagation. Afterwards, the supernatants containing the viruses were harvested as recombinant viruses.

### 2.6. Syncytium Assay for Cell Fusion

The syncytium assay was performed as previously described [21]. We co-transfected A549 cells with either wild-type or mutant HA-expressing pCAGGS-based plasmids and a green fluorescent protein (Venus)-expression plasmid (pcDNA3.1-Venus), using polyethylenimine (PEI; Polysciences, Warrington, PA, USA). After transfection, the cells were incubated at 37 °C for 24 h. Afterwards, we washed the cells twice with phosphate-buffered saline containing Mg^2+^ and Ca^2+^ (PBS+). Next, the cells were treated with 5 µg/mL TPCK-trypsin and incubated for 5 min at 37 °C. We used PBS+ containing 4% NCS to inactivate the trypsin. To induce cell fusion, we treated the cells with pH-adjusted PBS+ with citric acid for 1 min and incubated them in DMEM with 10% FBS at 37 °C for 1 h. A fluorescence microscope (Axio Vert.A1; ZEISS, Oberkochen, Germany) was used to visualize the fused cells.

### 2.7. Thermostability Assay

The thermostability assay was performed as previously described [22]. We diluted either wild-type or mutant viruses to 1 × 10^5^ PFU/mL and dispensed them into 5 aliquots of 200 µL. All of the aliquots were heated at 50 °C for 0, 30, 60, 90, or 120 min and immediately placed on ice. The virus titer was measured using a plaque assay.

### 2.8. Virus Release Assay

To assess the HA–NA functional balance of the viruses, a virus release assay was performed on erythrocytes [22]. Aliquots of 50 μL, which were prepared from 2-fold dilutions of virus-containing HA titers of 1:1024, were added to 50 μL of 0.7% turkey erythrocytes in a 96-well microplate and subsequently incubated at 4 °C for 1 h. The plate was then stored at 37 °C, and the decrease in HA titers was recorded at intervals.

### 2.9. Statistical Analysis

We computed the area under the curve (AUC) to compare the growth curves of the viruses. The statistical significance of the differences in AUC between viruses was analyzed using analysis of variance with post hoc Dunnett’s multiple comparisons test.

## 3. Results

### 3.1. Adaptation of H7N2 Feline Influenza Virus to A549 Cells

To adapt H7N2 feline IAV to human respiratory cells, we serially passaged it in A549 cells and obtained passaged viruses that grew efficiently. We observed clear cytopathic effects with extensive cell rounding at the 15th passage of the virus, so that the 16th passage virus was used as A549-adapted viruses, referred to as Aad-1, Aad-2, and Aad-3, for further analyses.

### 3.2. Replication of Aad Viruses in A549 Cells

To examine the replication properties of Aad viruses in A549 cells, we compared their growth kinetics with those of the wild-type (WT) virus at 37 °C (Figure 1). All of the Aad viruses showed significantly higher growth than the WT virus; approximately 1000-fold higher viral titers were measured at any time point post infection, suggesting that they might be adapted to A549 cells.

### 3.3. Sequence Analysis of A549-Adapted Viruses

We determined and compared the full-genome sequences of WT and Aad viruses, detecting different nucleotides in the HA, nucleoprotein (NP), and neuraminidase (NA) segments between WT and Aad viruses. All of them defined synonymous mutations, resulting in amino acid substitutions or deletions found at six positions on HA, one on NP, and three on NA: HA1-H16Q, HA1-N199S, and NA-I28S for Aad-1; HA1-G99S, HA2-I47T, HA2-Y119H, NA-deletion of TLH at positions 3436 (NA-Δ3436), and NA-V51A for Aad-2; and HA1-E105K, HA1-N199S, HA2-I47T, NP-E294G, and NA-I28L for Aad-3 (Table 1). There was no sequence difference in the PB2, PB1, PA, M, and NS segments between the WT and Aad viruses. We searched and mapped the amino acid position of each HA mutation using a protein databank for the 3D structure; HA1-G99S, HA1-E105K, and HA1-N199S were located in the head region of HA, and HA1-H16Q, HA2-I47T, and HA2-Y119H were located in the stalk region (Figure 2). In addition, among the NA mutations, NA-I28S/L and NA-Δ34 36 were located in the transmembrane region of NA, according to a previous study [23].

### 3.4. Generation and Replication of Recombinant Aad Viruses

Aad viruses, obtained by successive passages without limiting dilutions in the cells, may contain a minor population of heterogeneous viruses, although the sequence analysis did not show any considerably mixed nucleotides, except for the WT sequence. Therefore, we generated each Aad homogenous virus or WT virus as a control using reverse genetics, namely rAad-1, rAad-2, rAad-3, or rWT. Their growth properties were examined and compared with those of the original Aad-1, Aad-2, and Aad-3 viruses (Figure 1), indicating equivalent growth rates of the recombinant Aad viruses to the original Aad viruses, which were significantly higher than those of rWT (Figure 3A). In addition, we investigated the growth properties of recombinant Aad viruses at 33 °C, which may reflect the temperature in the human upper respiratory tract. There was no statistically significant difference in growth rates between 33 °C and 37 °C (Figure 3B), suggesting that they would replicate well in the temperature environment of the human respiratory organs.

### 3.5. Effects of Mutations on Virus Growth in A549 Cells

To identify the mutation(s) responsible for the high growth properties of Aad viruses in A549 cells, we generated recombinant viruses possessing a single mutation by reverse genetics. First, to test mutations found in Aad-1, when we compared the growth rates of rHA1-H16Q, rHA1-N199S, or rNA-I28S with that of rWT, all mutant viruses grew significantly better than rWT to various extents (Figure 4A). Among them, the HA1-H16Q mutation displayed the strongest effect on growth enhancement, albeit to a slightly lower level than rAad-1. To test the mutations found in Aad-2, when we compared the growth rate of rHA1-G99S, rHA2-I47T, rHA2-Y119H, rNA-Δ3436, or rNA-V51A viruses with that of rWT, the HA2-Y119H mutation showed the strongest effect and then NA-Δ3436 or HA2-I47T mutations possessed notable effects in growth enhancement (Figure 4B). The HA1-G99S mutation had no effect, whereas NA-V51A had a negative effect on the growth rate. To test the mutations found in Aad-3, we compared the growth rates of rHA1-E105K, rNP-E294G, and rNA-I28L, as well as HA2-N199S and HA2-I47T, which were also found in other Aad viruses, with that of rWT, indicating that the NA-I28L mutation possessed an enhanced effect similar to that of HA2-I47T (Figure 4C) and NA-I28S found in Aad-1 (Figure 4A). NP-E294G had a negative effect on the growth rate. Taken together, these data indicate that no single mutation alone enhanced the growth to equivalent levels of the Aad viruses, but suggest that mutations in the HA stalk region and those in the NA transmembrane region could synergistically lead to the high growth of the H7 feline virus in human respiratory A549 cells.

### 3.6. Change in the pH Threshold of Syncytium Formation by HA Mutations

To evaluate the involvement of the HA1-H16Q, HA2-I47T, and HA2-Y119H mutations in the HA stalk region in its membrane fusion activity [24], we conducted a syncytium assay to determine the pH threshold of each HA mutant for cell fusion. In contrast, cell fusion was observed at pH 5.4 in wild-type HA, at pH 5.8 in HA1-H16Q or HA2-I47T mutants, and at pH 5.6 in the HA2-Y119H mutant (Figure 5). These data indicate that changes in the pH threshold for the cell fusion activity of HA were caused by these mutations, possibly leading to the high growth of the mutant viruses in A549 cells through adaptation to the endosomal pH environment in A549 cells.

### 3.7. Thermostability of HA Mutants

It is known that changes in the thermostability of the virus can be related to the adaptation of avian IAVs to humans, and it may affect the pH threshold for the membrane fusion activity of HA [25]. Because HA stability has been reported to be substantially affected by mutations in the stalk region [25], we examined the thermostability of HA mutant viruses such as rHA1-H16Q, rHA2-I47T, and rHA2-Y119H. Each aliquot containing 1 × 10^5^ PFU/mL was incubated at 50 °C for 0, 30, 60, 90, and 120 min, and changes in the virus titers were monitored (Figure 6). The titers of all the mutant viruses decreased significantly after 60 min of incubation. Among them, a greater reduction in titers was observed for rHA1-H16Q than for the others. These data suggest that changes in the thermostability of the virus due to mutations in the HA stalk region may be correlated with the enhanced growth of the mutant viruses in A549 cells.

### 3.8. Change in HA–NA Functional Balance by NA Mutations

The NA-I28S/L and NA-Δ3436 mutations significantly enhanced the growth of the mutant viruses (Figure 4B). These mutations may affect NA activity through undefined structural alterations, suggesting that the HA–NA functional balance may be altered. To assess this idea, we performed a virus release assay from erythrocytes and found a delayed release of the mutant viruses compared with the rWT virus (Figure 7), suggesting that the NA activity was affected by these mutations. In particular, it has been suggested that deletion mutations (Δ3436) highly reduce the NA activity of the virus. These findings suggest that the HA–NA functional balance of the viruses was altered by these mutations in the NA transmembrane region, leading to enhanced growth of mutant viruses in A549 cells.

## 4. Discussion

In the incidents reported so far, avian IAVs have been directly transmitted from infected poultry to humans, leading to disease and death of infected persons [26]. However, there has been no pandemic as a result of this situation, possibly because of the lack of emergence of mutant viruses that could be efficiently transmitted among human populations. Historically, pandemic viruses have emerged via pigs as an intermediate host by mutations and reassortments with avian, swine, and human viruses, because pigs are susceptible to both avian and human viruses [2,27,28]. Cats are also susceptible to both viruses [8,9,10,11,12,13,14,15] and have close contact with humans as companion animals, providing the potential to act as an intermediate host for the emergence of a novel pandemic virus. In this study, to gain insight into this potential, we investigated the molecular basis for the adaptation of avian-derived H7N2 feline IAV to human respiratory A549 cells. We found that mutations in both the HA stalk and NA transmembrane regions were able to synergistically enhance the growth rates of H7N2 feline IAV in A549 cells, suggesting its potential to sporadically evolve into a human pathogen, causing a pandemic, by adapting to human respiratory cells.

Here, we found that each Aad virus contained mutations concomitantly in both HA and NA proteins, although no specific mutation was common among the three Aad viruses (Table 1). The HA mutations detected were classified into two groups: one located in the head region and the other in the stalk region (Figure 2). The mutant viruses with the former (HA1-G99S, HA1-E105K, or HA1-N199S) grew to similar or slightly increased levels compared with rWT, indicating that these mutations were not responsible or were weakly responsible for adaptation to A549 cells. The HA head region of IAV is involved in receptor binding; avian virus HA preferentially binds to α2,3-linked sialic acid, whereas human virus HA binds to α2,6-linked sialic acid [29,30,31]. Therefore, changes in receptor specificity are among the most important determinants of avian virus adaptation to humans [2]. Both types of sialic acid are found in the respiratory epithelial cells of cats [32]. A previous study has shown that H7N2 feline IAV preferentially binds to α2,3-linked sialic acid through glycan array analysis [18]. Although an HA 3D structure predicted that, among these three mutations, HA1-N199S may affect receptor binding because of its location in the head region (Figure 2), weak enhancement in the growth rate indicated that receptor specificity should not be altered. This fact suggests that the sialic acid binding specificity of the Aad viruses might not be altered by mutations. Therefore, it is concluded that alteration of the receptor specificity is not mandatory for feline IAV to adapt to A549 cells. However, previous reports have shown that A549 cells contain abundant α2,3-linked sialic acid [33,34], suggesting a different receptor distribution from human upper respiratory cells, which mainly contain α2,6-linked sialic acid [35]. Thus, the findings of this study cannot be directly extrapolated to an adaptation mechanism for the human population in natural settings. Further investigation using primary human respiratory cells will provide clues to discuss this point.

In contrast, the recombinant viruses with HA1-H16Q, HA2-I47T, and HA2-Y119H in the HA stalk region grew much better than the rWT. In particular, rHA1-H16Q and rHA2-Y119H were replicated at nearly equivalent levels as the Aad viruses (Figure 4), suggesting that these mutations are strongly responsible for adaptation to A549 cells. Therefore, the alteration of the pH threshold for the membrane fusion activity could be a key determinant for the adaptation of feline viruses to humans, as supported by the previous notion that such pH requirements for membrane fusion can be one of the determinants of the host specificity of IAVs [25]. Interestingly, increases in the pH threshold for membrane fusion by these HA mutations might be related to the reduced thermostability of the mutant viruses (Figure 6), in agreement with the previous observation in the adaptation study of H5N1 avian IAV to human airway cells [36].

The NA mutations, NA-I28S/L and NA-Δ3436, that enhanced growth were detected in Aad viruses located in the transmembrane region. The virus release assay demonstrated that these mutations altered the HA–NA functional balance, which is regulated in combination with the receptor-binding activity of HA and the receptor-destroying activity of NA. The HA–NA functional balance is important for the growth, pathogenicity, and host range of IAVs [2,3,37,38,39]. Considering that HA1-G99S, HA1-E105K, and HA1-N199S mutations in the HA head region might not alter receptor-binding properties, as discussed above, it was confirmed that alteration of the HA–NA functional balance was achieved through the reduced NA activity by these mutations. The NA transmembrane region may act as a signal for association with lipid rafts and translocation from the endoplasmic reticulum to the apical surface, affecting the budding process of the virus [40,41,42]. Hence, the reduction in NA activity with NA-I28S/L or NA-Δ3436 mutations may be caused by alteration of the NA expression levels on the virions, leading to a high growth of Aad viruses in A549 cells by altering the HA–NA functional balance.

In conclusion, we identified HA and NA mutations that could lead to the enhanced growth of H7N2 feline IAV in human respiratory cells. The amino acids detected in the Aad virus HAs (i.e., HA1-16Q, HA2-47T, and HA2-119H) were not found in any H7 human isolates in the database, including Chinese H7N9 viruses that emerged in 2013–2017, and those in the NAs (i.e., NA-28S/L) were not found in seasonal H3N2 human viruses in 2016–2022. Exceptionally, only two H7N2 avian isolates of 467 registered sequences possessed NA-28L. Nonetheless, these mutations caused changes in the pH threshold for membrane fusion and the HA–NA functional balance, resulting in adaptation of the feline virus to A549 cells. We conclude that the H7N2 feline IAV has the potential to become a human pathogen by adapting to human respiratory cells, owing to the synergistic biological effect of mutations in the envelope glycoproteins. Our findings may contribute to the prediction and assessment of the risk of animal viruses for the emergence of a pandemic virus.

## Figures and Tables

**Figure 1 viruses-14-01091-f001:**
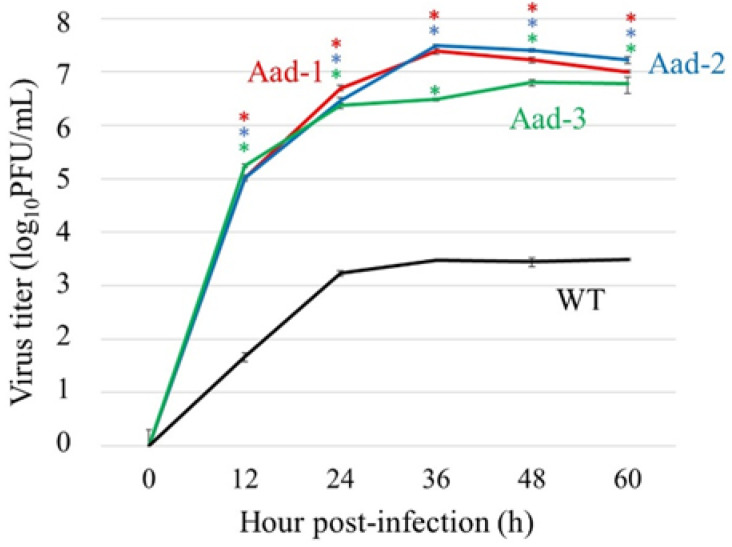
Growth kinetics of A549-adapted viruses in A549 cells. The cells were infected with A549-adapted (Aad-1, Aad-2, and Aad-3) or wild-type (WT) viruses at a multiplicity of infection of 0.01 at 37 °C. Supernatants were harvested at the indicated time points. Virus titers were determined using a plaque assay in MDCK cells. Data present the means ± standard deviation (SD) of the results of three independent experiments. * *p* < 0.05, compared with that of the WT virus using Dunnett’s test.

**Figure 2 viruses-14-01091-f002:**
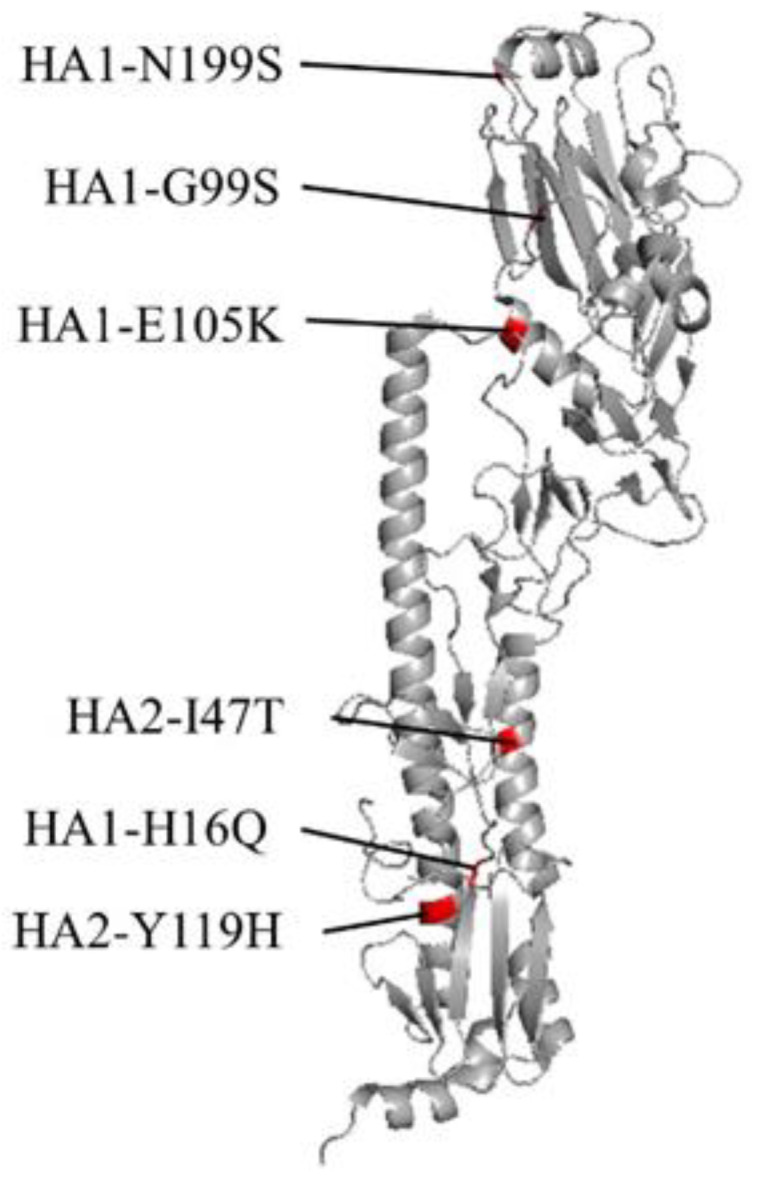
Predicted 3D structure of the H7-hemagglutinin (HA) protein. The structure of monomeric HA from A/environment/New York/30732-1/2005(H7N2) (PDB: 3M5J) is shown. Mutations found in A549-adapted viruses are colored in red.

**Figure 3 viruses-14-01091-f003:**
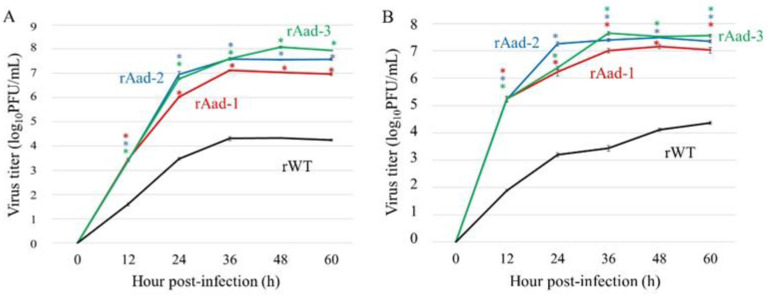
Growth kinetics of A549-adapted viruses in A549 cells at different temperatures. The cells were infected with recombinant A549-adapted viruses (rAad-1, rAad-2, and rAad-3) or recombinant wild-type (rWT) at a multiplicity of infection of 0.01 and incubated at 37 °C (**A**) and 33 °C (**B**). Supernatants were harvested at the indicated time points. Virus titers were determined using plaque assays in MDCK cells. Data present the means ± SD of the result of three independent experiments. * *p* < 0.05, compared with that of rWT using Dunnett’s test.

**Figure 4 viruses-14-01091-f004:**
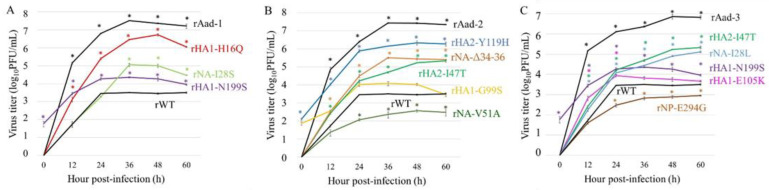
Growth kinetics of the mutant viruses in A549 cells. The cells were infected with recombinant wild-type virus (rWT), A549-adapted viruses, or mutant viruses possessing each mutation at a multiplicity of infection of 0.01 at 37 °C. (**A**) The growth of the mutant viruses found in Aad-1 (rHA1-H16Q, rHA1-N199S, or rNA-I28S), rAad-1, and rWT; (**B**) the growth of the mutant viruses found in Aad-2 (rHA1-G99S, rHA2-I47T, rHA2-Y119H, rNA-Δ3436, or rNA-V51A), rAad-2, and rWT; and (**C**) the growth of the mutant viruses found in Aad-3 (rHA1-E105K, rHA1-N199S, rHA2-I47T, rNP-E294G, or rNA-I28L), rAad-3, and rWT were included. Supernatants were harvested at the indicated time points. Virus titers were determined by plaque assays in MDCK cells. Data present the means ± SD of the results of three independent experiments. * *p* < 0.05, compared with that of rWT using Dunnett’s test.

**Figure 5 viruses-14-01091-f005:**
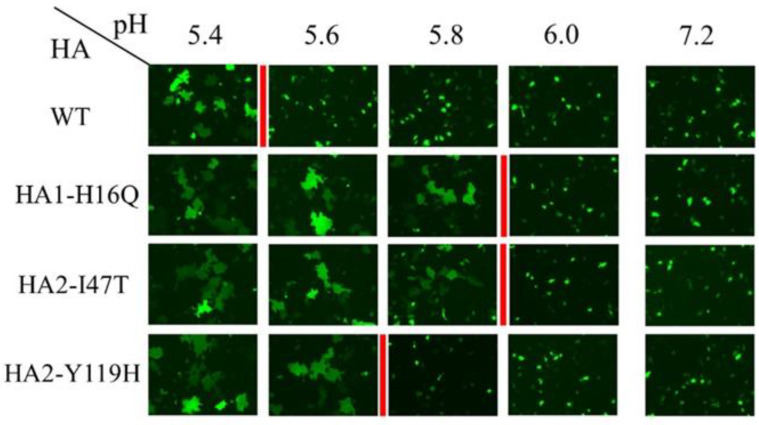
pH threshold for syncytium formation with HA mutations in A549 cells. The cells were co-transfected with the plasmids expressing mutant HAs (HA1-H16Q, HA2-I47T, or HA2-Y119H) or WT HA and a Venus-expression plasmid. Syncytium formation was induced with an adjusted pH buffer of pH 5.4 to pH 6.0 and with pH 7.2 as the control. The red line marks the border of pH for syncytium formation. The experiments were repeated thrice independently and the same results were obtained.

**Figure 6 viruses-14-01091-f006:**
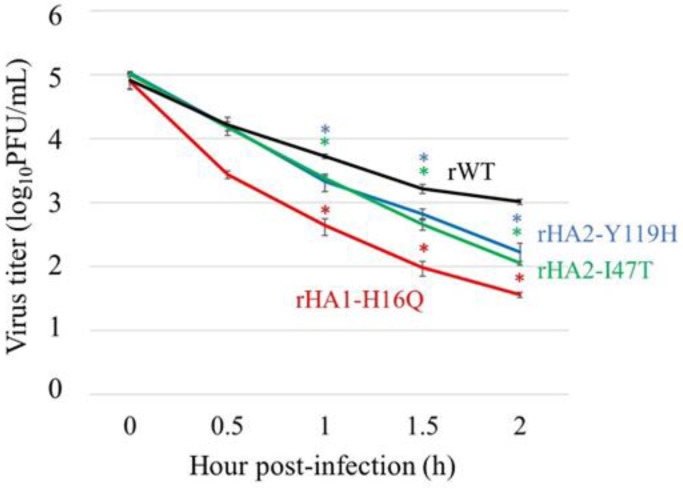
Thermostability of HA mutant viruses. Aliquots containing 1 × 10^5^ PFU/mL of the recombinant viruses possessing mutations (rHA1-H16Q, rHA2-I47T, or rHA2-Y119H) or rWT were heated at 50 °C for 0, 30, 60, 90, and 120 min, and immediately placed on ice. Virus titer was measured using plaque assay in MDCK cells. Each data point is the mean ± SD of triplicated experiments. * *p* < 0.05, compared with that of rWT using Dunnett’s test.

**Figure 7 viruses-14-01091-f007:**
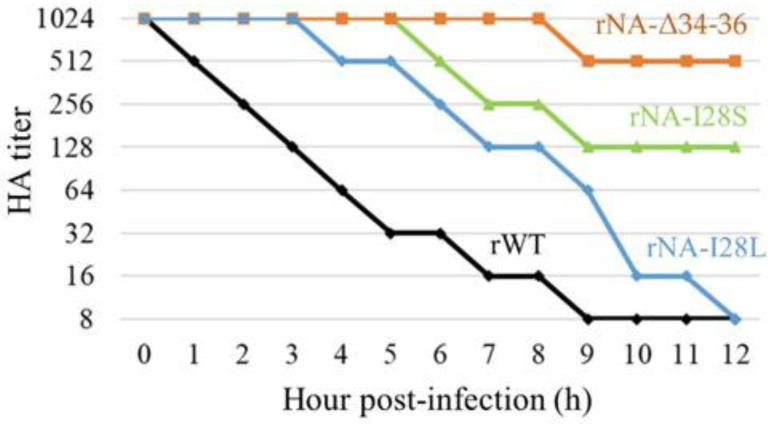
Elution of NA mutant viruses from erythrocytes. Two-fold dilutions of the NA transmembrane mutant rNA-I28S, rNA-I28L, or rNA-Δ3436 or rWT containing HA titers of 1:1024 were incubated with an equal volume of 0.7% turkey erythrocytes in microplate wells at 4 °C for 1 h. The plates were then stored at 37 °C, and the reduction in HA titers was evaluated periodically for 12 h.

**Table 1 viruses-14-01091-t001:** Amino acid substitutions found in A549-adapted viruses.

Virus	HA1				HA2		NP	NA				
	16 ^1^	99	105	199	47	119	294	28	34	35	36	51
Wild-type	H	G	E	N	I	Y	E	I	T	L	H	V
Aad-1	Q ^2^	G	E	S	I	Y	E	S	T	L	H	V
Aad-2	H	S	E	N	T	H	E	I	– ^3^	–	–	A
Aad-3	H	G	K	S	T	Y	G	L	T	L	H	V

^1^ Amino acid position. ^2^ Amino acid different from wild-type colored red. ^3^ –, deletion.

## Data Availability

Not applicable.

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
