# Peer review of "Adaptation of the H7N2 Feline Influenza Virus to Human Respiratory Cell Culture"

_viruses, 2022, doi:10.3390/v14051091_

Round 1

Reviewer 1 Report

The authors present very interesting and relevant data on possible mutations responsible for increased growth and adaptation of feline influenza viruses to human respiratory cells. The findings are clearly presented. Minor comments for consideration are listed below:

Line 24. Please can the author clarify what is meant by ‘human agent’? Human pathogen may be more appropriate. Also mentioned in the discussion on lines 291 and 342

Line 29. ‘…are endemic to humans and other mammals’ should be ‘…endemic in humans and other mammals’

Line 29. Please cite a more appropriate and recent reference to support the statement

Line 32. respiratory signs rather than respiratory symptoms is more appropriate

Line 35. Change ‘low pathogenic’ to ‘low pathogenicity’ here and throughout the manuscript

Line 40. Change ‘highly pathogenic’ to ‘high pathogenicity’

Line 42. Please clarify the statement as it appears to contradict the previous sentences about reports of IAV transmission from cats to humans. ‘Although transmission of IAV from companion animals to humans has not been reported ‘

Line 47. please clarify what is meant by ‘In the consensus feature’

Line 53. Change ‘large number of infected persons and the death in China’ to ‘large number of human infections and deaths in China’.

Line 188: ‘no appreciable difference’. Does the author mean there was no statistical significance?

Discussion.  Can the author include a comment on whether any of the mutations obtained after passaging in A549 cells are seen in nature e.g. in animal and human virus sequences on public databases or described in the literature. Were there any sequence differences identified between the cat and human cases in New York during 2016-2017?

Author Response

Responses to Reviewer 1

“Line 24. Please can the author clarify what is meant by ‘human agent’? Human pathogen may be more appropriate. Also mentioned in the discussion on lines 291 and 342”

We altered the words in the revised manuscript (lines 24, 303, and 362).

“Line 29. ‘…are endemic to humans and other mammals’ should be ‘…endemic in humans and other mammals’”

We corrected the word in the revised manuscript (line 29).

“Line 29. Please cite a more appropriate and recent reference to support the statement”

We replaced ref. [1] to a recent reference in the revised manuscript (see lines 384-385).

“Line 32. respiratory signs rather than respiratory symptoms is more appropriate”

We altered the word in the revised manuscript (line 33).

“Line 35. Change ‘low pathogenic’ to ‘low pathogenicity’ here and throughout the manuscript”

We corrected the words throughout the revised manuscript (lines 35, 48, and 52).

“Line 40. Change ‘highly pathogenic’ to ‘high pathogenicity’”

We corrected the words throughout the revised manuscript (lines 40, 48, and 52).

“Line 42. Please clarify the statement as it appears to contradict the previous sentences about reports of IAV transmission from cats to humans. ‘Although transmission of IAV from companion animals to humans has not been reported ‘”

We agreed with this point and clarified the statement in the revised manuscript (line 43).

“Line 47. please clarify what is meant by ‘In the consensus feature’”

We corrected the word in the revised manuscript (line 47).

“Line 53. Change ‘large number of infected persons and the death in China’ to ‘large number of human infections and deaths in China’.”

We corrected the phrase in the revised manuscript (lines 53-54).

“Line 188: ‘no appreciable difference’. Does the author mean there was no statistical significance?”

We corrected the phrase in the revised manuscript (lines 199-200).

“Discussion.  Can the author include a comment on whether any of the mutations obtained after passaging in A549 cells are seen in nature e.g. in animal and human virus sequences on public databases or described in the literature. Were there any sequence differences identified between the cat and human cases in New York during 2016-2017?”

We included this information in the last paragraph of the revised manuscript (lines 354-359) as follows; “The amino acids detected in the Aad virus HAs (i.e., HA1-16Q, HA2-47T, and HA2-119H) were not found in any H7 human isolates in database, including Chinese H7N9 viruses emerged in 2013–2017, and those in the NAs (i.e., NA-28S/L) were not found in seasonal H3N2 human viruses in 2016–2022. Exceptionally, only two H7N2 avian isolates of 467 registered sequences possessed NA-28L.”.

Reviewer 2 Report

This manuscript describes some mutations of HA and NA protein from a H7N2 avian influenza virus isolated from feline could enhance the replication in human respiratory culture cells. A H7N2 feline IAV ( A/feline/New York/WVDL-14/2016 (H7N2)) was serially passaged in human respiratory A549 cells and adaptation viruses that grew efficiently were obtained. Some amino acid substitutions or deletions on HA,NA and NP of three A549-adapted viruses were observed with full-genome sequences analysis. Using reverse genetics, the authors finally confirmed that some mutations could enhance viral growth in A549 cells, including three mutations (HA1-H16Q, HA2-I47T, or HA2-Y119H) in HA and one I28S/L mutation /3 amino acid deletion in NA. This is largely a research work which in itself offers new information regarding the potential cross-species infection human for animal influenza virus.

There is a minor point that should be addressed:

(1) How about the sialic acids-bind specificity(α-2,3- or α-2,6-linkages)for both wild virus and A549-adapted viruses?

Author Response

Response to Reviewer 2

“(1) How about the sialic acids-bind specificity (α-2,3- or α-2,6-linkages) for both wild virus and A549-adapted viruses?”

Previous study has shown that H7N2 feline IAV preferentially binds to α2,3-linked sialic acid by glycan array analysis [18: Hatta et al., Emerg Infect Dis. 2018, 24, 75-86]. Although we did not assess the sialic acids-binding specificity for our Aad viruses, we discussed that it might not be altered by mutations, because the growth rate of the recombinant virus with each HA1 mutation was not significantly or dramatically changed. This fact suggests that the sialic acids-binding specificity of the Aad viruses might not be altered by mutations. We added this point in the Discussion of the revised manuscript (lines 315-317 and 320-321).